# Lake Cyanobacterial Bloom Color Recognition and Spatiotemporal Monitoring with Google Earth Engine and the Forel-Ule Index

**Ting Song [1], Ge Liu [2], Hujun Zhang [1], Fei Yan [1], Yingbo Fu [3] and Junyi Zhang [1,*]**

[1] Jiangsu Wuxi Environmental Monitoring Center, Wuxi 214121, China; songting2211@163.com (T.S.)
[2] The Northeast Institute of Geography and Agricultural Ecology, Changchun 130102, China; liuge@iga.ac.cn
[3] School of Cybersecurity, Northwestern Polytechnical University, Xi'an 710072, China
[*] Correspondence: blocksharon@163.com

**Abstract:** Cyanobacterial blooms represent a significant environmental problem, threatening aquatic ecosystems worldwide. Caused by the eutrophication of water bodies and global climate change, these blooms have altered freshwater ecosystems worldwide during recent decades. Although cyanobacterial blooms are typically caused by blue-green cyanobacteria, which derive their color from the phycocyanin pigment, other pigmented cyanobacterial blooms have been frequently observed in water bodies. These blooms pose a serious environmental threat to inland waters, endangering global public health and aquatic ecosystems. Therefore, comprehending the mechanism of color variation in cyanobacterial blooms is crucial for revealing the outbreak mechanism and implementing effective prevention and control measures. This study developed a human–machine interactive workflow for extracting cyanobacterial blooms and recognizing their colors based on the Forel-Ule index and Sentinel-2 MultiSpectral Instrument data. Using this workflow, the authors conducted spatiotemporal analysis and statistical analysis of bloom color for cyanobacterial blooms in four typical eutrophic lakes from 2019 to 2022. The findings indicated a declining trend in cyanobacterial blooms across the four studied lakes over the years, among which Hulun Lake experienced an annual increase in cyanobacterial blooms and emerged as the lake with the most severe outbreak of such blooms in 2022. The yellowing status of cyanobacterial blooms varied among the four lakes, with Taihu Lake and Dianchi Lake exhibiting a relatively high proportion of green-yellow and yellow cyanobacterial blooms, followed by Chaohu Lake, whereas Hulun Lake had the lowest occurrence. The workflow developed in this study was implemented in Google Earth Engine and provided an automated, integrated, and rapid monitoring solution for the long-term monitoring and color recognition of cyanobacterial blooms.

**Keywords:** cyanobacteria blooms; Google Earth Engine; cloud computing; Forel-Ule index

## 1. Introduction

Cyanobacterial blooms occur when water bodies that are rich in nitrogen and phosphorus experience abundant growth of cyanobacteria under suitable environmental conditions, such as temperature, light, and wind speed. These blooms may have deleterious effects on the ecological environment of the water body. These cyanobacteria often aggregate into visible clusters, resulting in a noticeable decrease in water column transparency. The coloration of algal cells serves as a comprehensive representation of their photosynthetic pigments. Algal photosynthetic pigments are a group of chemical substances that combine with proteins to form pigment proteins in photosynthetic organs. They can be categorized into three categories according to their molecular structures: chlorophylls, carotenoids, and phycobiliproteins. During photosynthesis, these pigments capture light energy and transform it into chemical energy. The primary pigments in cyanobacterial cells are chlorophyll, phycocyanin, and phycoerythrin. These pigments absorb less blue-green light, resulting in a predominant

blue-green appearance [1]. This is the reason why cyanobacterial blooms are characterized by blue-green hues. With the advancement of monitoring and research on cyanobacterial blooms, yellow-green and yellow cyanobacterial blooms have been observed in various bodies of water. For example, in October 2012, Cai et al. reported a noteworthy example, where the cyanobacterial blooms dominated by *Microcystis* in Lake Taihu underwent a transition from blue-green to yellow during July [2]. In 2000, Lake Rotoehu, a eutrophic lake in New Zealand, experienced a significant presence of *Microcystis*, resulting in the water turning mustard yellow. While alternative hypotheses suggest that this color change may have been attributed to different types of algae, the predominant presence of yellow *Microcystis* in live samples supports this observation. This phenomenon has been observed to persist over an extended period of time and has also been documented in Spain, where cyanobacterial blooms dominated by the cyanobacterium *Woronichinia naegliana* of the phylum Cyanobacteria made the water turn yellow. In addition, Lothar et al. observed yellow cyanobacterial blooms dominated by *Anabaena* in Lake Victoria [3].

Cyanobacterial blooms are one of the most serious environmental problems faced by inland waters, posing a significant threat to global public health and aquatic ecosystems [4]. The rapid economic development in China has led to the prevalent issue of eutrophication in freshwater lakes, resulting in frequent occurrences of algal blooms. Not only do these blooms affect the appearance of lakes, but the cyanotoxins they produce can also directly impact the health of humans and animals [5]. Under specific conditions, high-density cyanobacterial blooms have the potential to reduce dissolved oxygen, resulting in water discoloration and foul odor [6,7]. Therefore, precise and efficient monitoring of their spatial and temporal distribution is essential for prevention and management [8]. Satellites offer the advantages of extensive coverage and high temporal resolution, facilitating long-term and dynamic monitoring of cyanobacterial blooms. The identification of cyanobacterial blooms based on remote sensing technology has been extensively investigated through single-band thresholding [9], Normalized Difference Vegetation Index (NDVI), Enhanced Vegetation Index (EVI) [10], Floating Algae Index (FAI) [11,12], among others. The identification is primarily based on the strong reflection of the green band in the presence of cyanobacterial blooms, resulting in their visible appearances as green, as well as the strong reflection on the near-infrared reflectance characteristics that exhibit red-edge features similar to those of terrestrial vegetation. However, the varying qualities of satellite data and the limited robustness of a threshold, coupled with interference from factors such as clouds, aquatic plants, and highly turbid water bodies, often result in unsatisfactory results in practice. Therefore, further research is necessary to improve the remote-sensing-based identification of cyanobacterial blooms. The application of the Forel-Ule index (FUI) for identifying cyanobacterial blooms has made significant progress in recent years. Hou [13] and Dai [14] used Landsat and MODIS satellite data to monitor global lake and coastal blooms, demonstrating strong generalization capabilities. The MultiSpectral Instrument (MSI) onboard the Sentinel-2 satellite has several advantages, including high spatial resolution (10 m), short revisit period (less than five days), and a wide range of spectral bands (13 spectral bands from visible, near-infrared to short-wave infrared). Therefore, the utilization of the FUI and Sentinel-2 MSI was deemed appropriate for monitoring cyanobacterial blooms in small- and medium-sized lakes.

Satellite imagery can be used to distinguish cyanobacterial blooms from other waterbodies by employing various spectral bands to generate distinct colors. For instance, cyanobacteria manifest as yellow-green in RGB true color and as bright green in SWIR/NIR/Red, facilitating their differentiations from other land cover types. This study used the B11 (1613 nm)–B8 (833 nm)–B2 (442 nm) band combination of the Sentinel-2 MSI to construct the FUI, which was then combined with the hue angle $\alpha$ to develop a decision tree for cyanobacterial blooms' detection. In addition, the B4 (665 nm)–B3 (559 nm)–B2 (442 nm) band combination of Sentinel-2 MSI was employed to construct the FUI and characterize, classify, and identify cyanobacterial blooms based on ground-truthed, yellow-colored cyanobacterial bloom spectra.

Traditional remote sensing methods encounter numerous challenges, including the need for large data downloads, extended processing times, and high storage require-

ments [15], which impede automatic remote sensing monitoring of cyanobacterial blooms over long time series. In large-scale satellite remote sensing data processing scenarios, a specialized cloud platform can perform better. The Google Earth Engine (GEE) is a cloud platform designed for geographic spatial data analysis. It offers an interactive web development environment that stores petabyte-level remote sensing data and products, encompassing MODIS, Landsat, and Sentinel [16]. Data's online accessibility, rather than traditional downloads and pre-processing locally, makes GEE especially suitable for long time series and large-scale remote sensing applications. This study aims to address the urgent need for long-term monitoring of cyanobacterial blooms in inland eutrophic lakes, with the help of the latest development in cloud computing technology. To achieve this goal, this study combined the FUI with ground-measured spectral data and Sentinel-2 MSI multispectral data. An interactive and automated workflow was then developed on the GEE platform, enabling the rapid extraction of cyanobacterial blooms in inland eutrophic lakes, as well as characterizing, classifying, and identifying cyanobacterial blooms based on their colors. Furthermore, the research investigated the temporal and spatial distribution patterns of cyanobacterial blooms in four representative eutrophic lakes in China. To gain deeper insights, an analysis was conducted to examine the distinct color characteristics exhibited by cyanobacterial blooms in each lake, integrating the measured nutrient data of the lakes with the color expressions of the blooms.

## 2. Data and Study Area

### 2.1. Study Area

Four eutrophic lakes with frequent cyanobacterial blooms were selected as representative study areas, i.e., Lake Hulun, Lake Chaohu, Lake Taihu, and Lake Dianchi. These eutrophic lakes, selected for the purpose of studying the occurrences of cyanobacterial blooms, exhibited variations in size, shape, and location, as depicted in Figure 1.

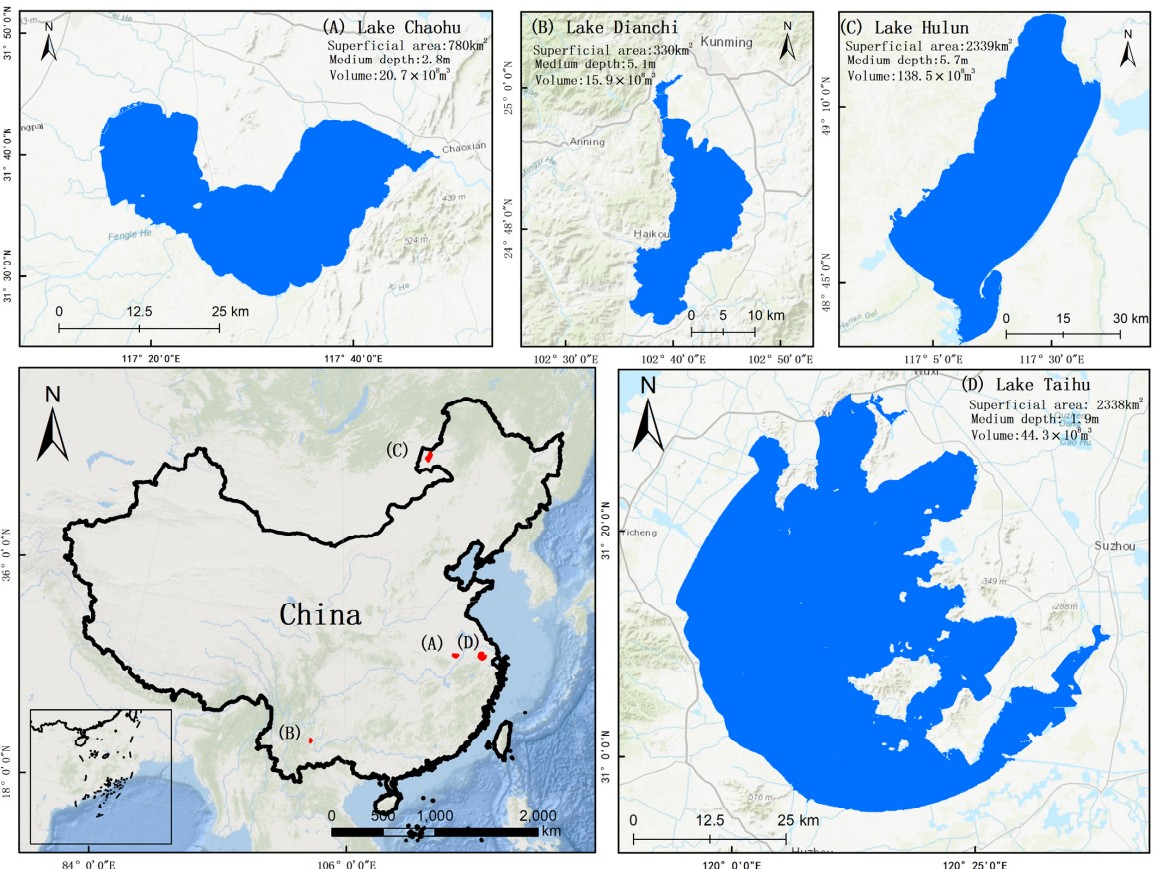

**Figure 1.** Locations of studied lakes.

### 2.2. Satellite Data

MSI Level-2A products were used in this study. They provided bottom-of-atmosphere corrected reflectance data, which had been radiometric calibrated and atmospheric corrected.

### 2.3. Field-Measured Data of Typical Yellow Cyanobacterial Blooms

On 28 July 2022, yellow cyanobacterial blooms emerged along the shore of Gonghu Bay in Taihu Lake. In response, two sampling points were set up to collect on-site samples for analysis.

On 2 August, two sampling points were established along the shore of the Meiliang Bay in Taihu Lake to collect green algal bloom control samples. In areas with prominent aggregations of cyanobacterial blooms, reflectance data from the water was captured using an ASD FieldSpec HandHeld 2 spectroradiometer. Additionally, water quality monitoring was conducted, and samples of algae were collected. Analysis revealed that *Microcystis* spp. was the dominant species among four cyanobacterial bloom samples, with a dominance exceeding 95%. Photos of the sampling sites are presented in Figure 2.

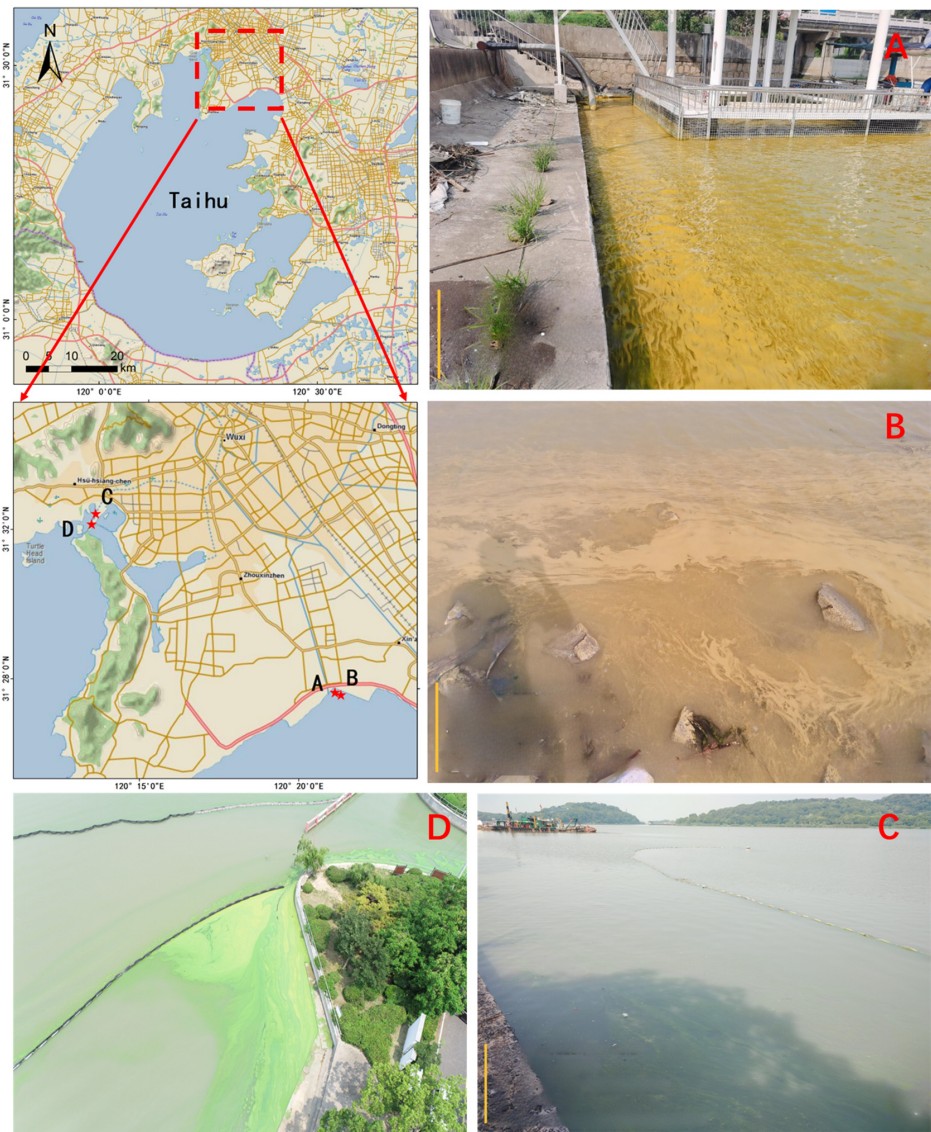

**Figure 2.** Cyanobacterial blooms caused by *Microcystis aeruginosa* ((**A**) in situ image 1 taken on 28 July 2022; (**B**) in situ image 2 taken on 28 July 2022; (**C**) in situ image 3 taken on 2 August 2022; (**D**) aerial image taken on 2 August 2022). The red stars represent the sampling positions on the map.

## 3. Methods

### 3.1. Methodology and Analysis Framework

A rapid monitoring and color recognition system for cyanobacterial blooms using Sentinel-2 MSI data was developed in this study, with the FUI serving as the primary method. This system enabled quick bloom detection and assessment of their growth statuses based on their colors. Figure 3 summarizes the overall process.

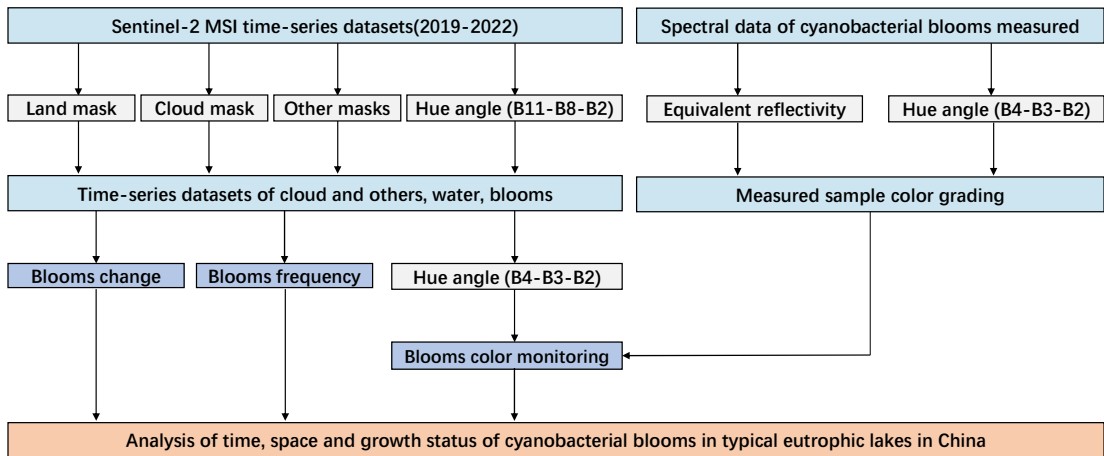

**Figure 3.** The analysis flowchart comprises an input dataset, remote sensing data preprocessing, automated extraction of cyanobacterial blooms, color recognition, spatiotemporal analysis of cyanobacterial blooms, and assessment of growth status.

First, the remote sensing data were preprocessed to mask land and clouds, followed by constructing a decision tree for the FUI using the B11 (1613 nm)—B8 (833 nm)—B2 (442 nm) band combination of Sentinel-2 MSI to extract cyanobacterial blooms. Then, combining the measured spectral data of ground samples, the FUI was reconstructed using the B4 (665 nm)—B3 (559 nm)—B2 (442 nm) band combination of Sentinel-2 MSI to characterize, grade, and identify the colors of cyanobacterial blooms. Finally, spatial and temporal analyses for cyanobacterial blooms were conducted on four representative eutrophic lakes in China, and the growth statuses of cyanobacterial blooms in these lakes was evaluated based on their color representations as well.

### 3.2. Processing of Satellite Data for Cyanobacterial Bloom Extraction

#### 3.2.1. Land Masking

In order to ensure accurate land masking and minimize interference and errors caused by terrestrial pixels, we used the latest Sentinel-2 MSI imagery to extract vector boundaries of water bodies in Taihu Lake, Hulun Lake, Chaohu Lake, and Dianchi Lake using the Modified Normalized Difference Water Index (MNDWI). Then, a 20 m inward buffer was applied to the extracted vector, mitigating the impact of mixed land–water boundary pixels. The MNDWI formula is as follows:

$$MNDWI = \left( R_{rs}(\text{green}) - R_{rs}(\text{swir}) \right) / \left( R_{rs}(\text{green}) + R_{rs}(\text{swir}) \right) \tag{1}$$

where $R_{rs}(\text{green})$ and $R_{rs}(\text{swir})$ represent the reflectances of the green and shortwave infrared bands, corresponding to Sentinel-2 MSI's B3 (560 nm) and B11 (1613 nm), respectively.

#### 3.2.2. Cloud Masking

This study employed a single-band threshold of $R_{rs}(664nm)$, which was greater than 0.2, as a cloud mask to filter out most cloud interference. However, this method failed to remove thin clouds and shadows effectively, leading to false positive results in cyanobacterial bloom extraction. To address this issue, we developed three indexes, namely, Index1,

Index2, and Index3, to eliminate false positives caused by thin clouds and shadows. The formulas for the indexes were as follows:

$$Index1 = (R_{rs}(green) - R_{rs}(blue)) / (R_{rs}(green) + R_{rs}(blue)) \qquad (2)$$

$$Index2 = (R_{rs}(rede) - R_{rs}(red)) / (R_{rs}(rede) + R_{rs}(red)) \qquad (3)$$

$$Index3 = (R_{rs}(green) - R_{rs}(red)) / (R_{rs}(green) + R_{rs}(red)) \qquad (4)$$

where $R_{rs}(blue)$, $R\_R_{rs}(green)$, $R_{rs}(red)$, and $R_{rs}(rede)$ denote the reflectances of the blue, green, red, and red-edge bands, respectively, corresponding to Sentinel-2 MSI's B1 (442 nm), B3 (559 nm), B4 (665 nm), and B5 (704 nm). Thresholds of Index1 (>0.1), Index2 (>0.15), and Index3 (>0.13) were determined through a trial-and-error approach based on visual assessment. Only the pixels that met all three criteria were classified as cyanobacterial blooms.

### 3.2.3. Mask Processing of Other Factors

Even after masking land and clouds, water surface reflectance data may still exhibit unusually high or low values. These values may be attributed to residual errors from the atmospheric correction method. Therefore, to exclude these pixels, several thresholds have been employed. Specifically, pixels with $R_{rs}(green)$ greater than 0.02 and all visible band reflectance values less than 0.5 are excluded [13]. Additionally, a turbidity index (TI) is used to mask high-turbidity water bodies. The formula for TI is

$$TI = (R_{rs}(red) - R_{rs}(green) - (R_{rs}(nir) - R_{rs}(green) \times 0.5) \qquad (5)$$

where $R_{rs}(green)$, $R_{rs}(red)$, and $R_{rs}(nir)$ correspond to Sentinel-2 MSI's B3 (559 nm), B4 (665 nm), and B8 (833 nm) bands, respectively. Considering that high turbidity water exhibited greater reflectance in the red band compared to the other two bands, a threshold of TI > 0 was established for TI [17].

### 3.2.4. Forel-Ule Index

The CIE 1931 XYZ color space, developed by the International Commission on Illumination (CIE) in 1931, defines the overall impression of color perceived by the human eye based on mathematical principles. In theory, the color parameters are calculated from three visible spectral bands (red, green, and blue) of satellite imagery and transformed into the CIE color space [18]. The goal of this color system is to simulate the integrated effect of the X, Y, and Z tristimulus values perceived by human vision. The transformation from RGB to X, Y, and Z can be expressed as follows:

$$\begin{aligned} X &= 2.7689 \times R + 1.7517 \times G + 1.1302 \times B \\ Y &= 1.0000 \times R + 4.5907 \times G + 0.0601 \times B \\ Z &= 0.0000 \times R + 0.0565 \times G + 5.5934 \times B \end{aligned} \qquad (6)$$

In this study, the B11 (1613 nm)–B8 (833 nm)–B2 (442 nm) band combination of Sentinel-2 MSI was assigned to the B, G, and R channels, respectively. The CIE chromaticity coordinates (x, y) were then determined by normalizing X, Y, and Z values to a range between 0 and 1. As x + y + z = 1, a specified color could be determined by the two values of x and y. Therefore, the CIE-xy chromaticity diagram (Figure 4) could be used to represent all colors within the visible range. The normalized simulated values x, y, and z were calculated using Equation (7):

$$x = \frac{X}{X+Y+Z}; y = \frac{Y}{X+Y+Z}; z = \frac{Z}{X+Y+Z} \qquad (7)$$

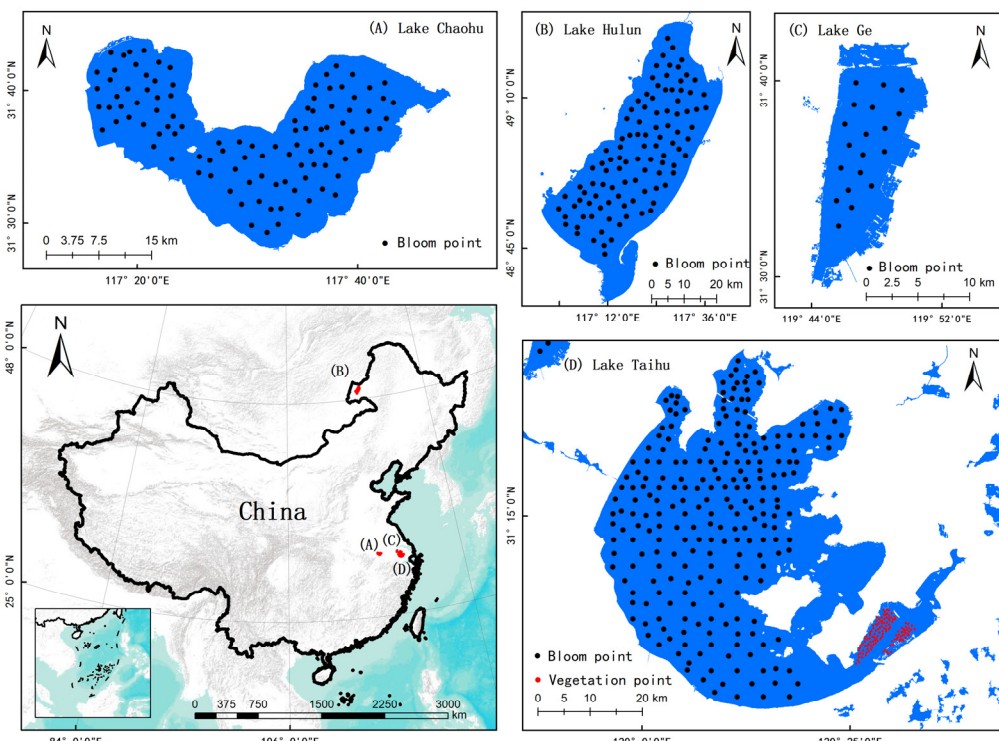

**Figure 4.** Hue Angle α Threshold Analysis Sample Point Distribution.

Finally, the hue angle α representing any pair of chromaticity coordinates (x′, y′) of the radiance spectrum could be calculated as follows:

$$\alpha = \left(\text{ARCTAN2}\frac{y'}{x'}\right) \times \frac{180}{\pi} = \left(\text{ARCTAN2}\frac{x - 0.333}{y - 0.333}\right) \times \frac{180}{\pi} \tag{8}$$

Here, ARCTAN2 is a four-quadrant arctangent function that allows the range of α to range from −180° to 180°. The increase in hue angle α indicates a transition from blue to red. In this study, the range of α was shifted from 0° to 360° by increasing it by180°.

### 3.2.5. Construction of the Decision Tree for Extracting FUI of Cyanobacterial Blooms

To determine the threshold of the hue angle α for cyanobacterial blooms, this study employed a random sampling method to collect 460 sample points from various water bodies, including 254 samples obtained from Lake Taihu and Lake Ge, 104 samples from Lake Chaohu, and 102 samples from Lake Hulun. The sampling excluded the eastern aquatic plant growth area in Lake Taihu. Additionally, 129 aquatic plant samples were collected using the same random sampling method to eliminate interference from cyanobacterial blooms. These samples were collected from the eastern part of Lake Taihu, where aquatic plants grow. The objective of collecting these samples was to study the hue angle α characteristics of aquatic plants (Figure 4).

First, a cloud mask was generated by applying a single-band threshold of $R_{rs}(664nm)$ > 0.2 to remove clouds. Then, the hue angle α and Floating Algae Index (FAI) values of the Sentinel-2 MSI remote sensing images from January 2019 to March 2023 were extracted from the bloom and aquatic plant samples. A sample was classified as either a bloom or an aquatic plant when its FAI value exceeded 0.04. The hue angle α of the sample was considered as the characteristic value of the bloom or aquatic plant. The FAI formula for Sentinel-2 MSI was as follows:

$$\begin{aligned} \text{FAI} &= R_{rs}(nir) - R'_{rs}(nir) \\ R'_{rs}(nir) &= R_{rs}(red) + (R_{rs}(swir) - R_{rs}(red)) \times \frac{833-664.5}{1613.7-664.4} \end{aligned} \tag{9}$$

where $R_{rs}(red)$, $R_{rs}(nir)$, and $R_{rs}(swir)$ denote the reflectances of the red, near-infrared, and shortwave infrared bands, respectively, corresponding to Sentinel-2 MSI B4 (664 nm), B8 (833 nm), and B11 (1613 nm). After data screening, 4214 bloom samples and 6149 aquatic plant samples were obtained, and the hue angle $\alpha$ histograms of both sample groups were statistically analyzed (Figure 5). In this histogram, the *x*-axis represented the hue angle $\alpha$, and the *y*-axis represented the frequency distribution of corresponding sample count.

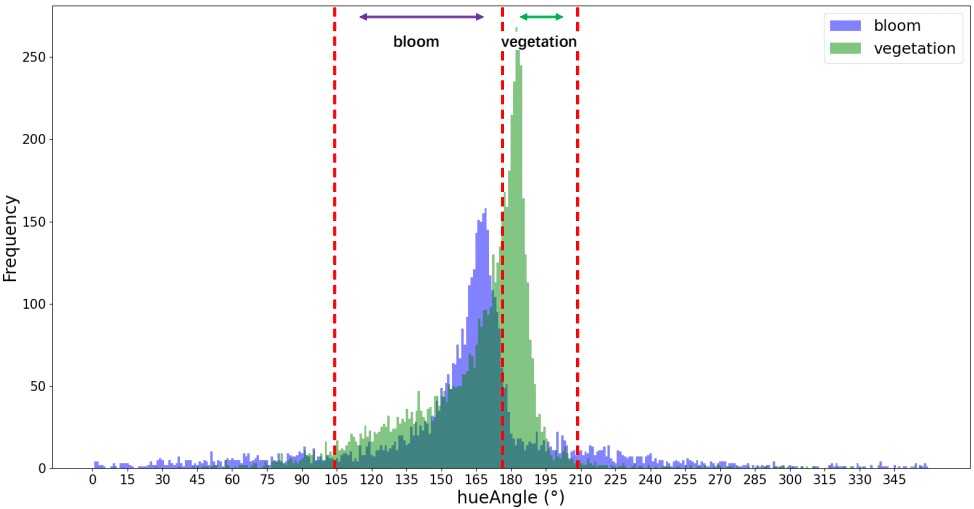

**Figure 5.** Threshold schematic diagram of FUI (SWIR/NIR/Red) based on the MSI sensor. The red dot lines represent the threshold segmentation range.

As depicted in Figure 5, the hue angle $\alpha$ histograms of aquatic plants and bloom samples exhibited discernible differences. According to the distribution results, the hue angle $\alpha$ range of $104° < \alpha < 179°$ was identified as the distribution interval for the cyanobacterial blooms and was classified as a bloom, whereas the hue angle $\alpha$ range of $179° \leq \alpha < 208°$ was identified as the distribution interval of aquatic plant hue angle $\alpha$ and was classified as clouds or other phenomena. Other hue angle $\alpha$ values were classified as water bodies, and the threshold and decision tree rules for extracting the cyanobacterial blooms' FUI values were constructed accordingly.

### 3.2.6. Time Series Analysis

To automatically merge data from the same day, the time attributes of Sentinel-2 MSI data were utilized. Next, cloud cover, water bodies, and cyanobacterial blooms were separately quantified based on classification results acquired from each satellite overflight image. This enabled daily area of cyanobacterial bloom to be obtained for Hulun Lake, Taihu Lake, Chaohu Lake, and Dianchi Lake. The annual average areas were calculated based on these data.

To compute the annual cumulative value of each pixel, the classification results obtained each time for cyanobacterial blooms and water bodies were binarized in each iteration, with every pixel being categorized as either a cyanobacterial bloom or a water body. The spatial distribution frequency index (SDFI) approach was employed to quantify the variations in the spatial distribution frequencies of cyanobacterial blooms across different regions within the study area. The formula used to calculate SDFI was:

$$\text{SDFI} = \frac{\sum_{i=1}^{n} R_{i\_bloom}}{\sum_{i=1}^{n} R_{i\_bloom} + \sum_{i=1}^{n} R_{i\_water}} \times 100\% \tag{10}$$

where $R_{i\_bloom}$ is the cyanobacterial bloom pixel value in the binary image of the i-th day, and $R_{i\_water}$ is the water body pixel value in the binary image of the i-th day. Finally, the annual frequency of cyanobacterial bloom occurrence in pixels was obtained, and the results were visualized.

### 3.3. Cyanobacterial Blooms Color Identification and Classification

3.3.1. Spectral Feature Analysis from Field Measurements

To analyze the spectral response characteristics of yellow and blue-green cyanobacterial blooms, ASD Field Spec Pro FR portable spectrometers were used to measure the spectra of yellow blooms #1 and #2 and blue-green control blooms #3 and #4 (Figure 6).

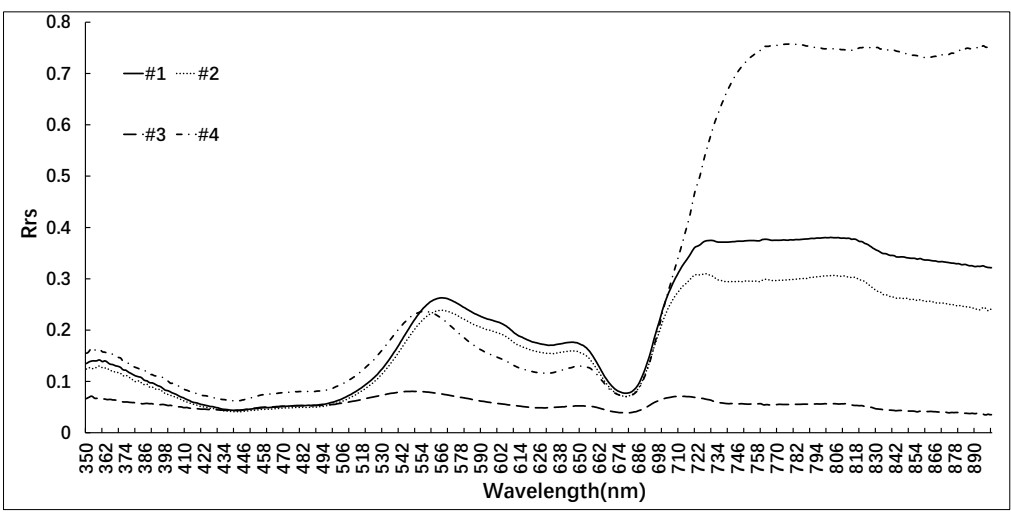

**Figure 6.** In situ measured cyanobacterial blooms of different colors: yellow blooms #1 and #2 and blue-green control blooms #3 and #4.

Figure 6 shows that 1#, 2#, and 4# exhibit very clear vegetation characteristics, whereas 3# displays spectral characteristics more akin to highly pigmented water bodies, which is consistent with the distribution of cyanobacterial blooms in the field. The low reflectance of water within the range of 400–500 nm can be attributed primarily to the strong absorption of chlorophyll a and yellow substances within this band of light energy. The weak absorption of chlorophyll and carotenoids, along with the scattering effect of cells, result in a reflection peak within the 550–580 nm range. Meanwhile, the absorption of phycocyanin at 620–630 nm creates a reflection trough, whereas the strong absorption of chlorophyll a in the red band near 670 nm also produces a reflection trough. Bloom 3# exhibits the fluorescence peak characteristics of chlorophyll in the 710 nm and beyond, with a rapid increase in reflectance at other wavelengths, indicating distinct vegetation features. The spectral characteristics of cyanobacterial blooms with different colors are primarily reflected in the central wavelength of the reflection peak within the range of 550–580 nm. The yellow cyanobacterial blooms 1# and 2# exhibit a reflection peak at a central wavelength of 566 nm, whereas the blue-green control bloom 3# displays a reflection peak at a central wavelength of 550 nm. The slightly yellowish control bloom 4#, on the other hand, reaches its reflection peak at a wavelength of 555 nm. Therefore, it can be inferred that blooms with more yellow colorations tend to exhibit a longer central wavelength position of the reflection peak in the 550–580 nm range, whereas those with greener colorations tend to show shorter central wavelength positions.

3.3.2. Identification of Cyanobacterial Blooms Based on Color

To meet the requirements of operational applications, this study used the FUI based on visible light to identify and classify cyanobacterial blooms by color. For Sentinel-2 MSI images, the R, G, and B channels corresponded to bands B4 (665 nm), B3 (559 nm), and B2 (442 nm), respectively. To evaluate whether the color grading of the hue angle α conformed to visual perception, the FUI was calculated for a typical yellow bloom sample spectrum.

First, the in situ measured spectral data were processed using the spectral response function of Sentinel-2 MSI to derive the equivalent reflectance of the corresponding channel as:

$$R_{rs}(band_i) = \frac{\int_{\lambda 1}^{\lambda 2} R_{rs}(\lambda)SRF(\lambda)d(\lambda)}{\int_{\lambda 1}^{\lambda 2} SRF(\lambda)d(\lambda)} \tag{11}$$

where $R_{rs}(band_i)$ is the equivalent reflectance of the i-th band of the satellite, $\lambda_1$ and $\lambda_2$ are the wavelength ranges of the band, $R_{rs}(\lambda)$ is the measured spectral reflectance, and $SRF(\lambda)$ is the spectral response rate at the wavelength $\lambda$. Then, according to Equations (5)–(7), the hue angle $\alpha$ was obtained for four ground samples.

Van der Woerd et al. found that the hue angle $\alpha$ calculated by the multi-spectral channel exhibited a deviation $\Delta\alpha$ of $-5°$ to $20°$, which was not entirely random and could be roughly described using a specific fifth-order equation for different satellite sensors [19]. For Sentinel-2 MSI, Equation (11) could be used for the approximate calculation of the hue angle deviation $\Delta\alpha$:

$$\Delta\alpha = -61.805a^5 + 257.86a^4 - 300.67a^3 + 40.595a^2 + 65.296a - 9.3398 \tag{12}$$

where $a$ is the hue angle $\alpha$ adjusted to be ranged from $0°$ to $360°$ and then divided by 100. Adding $\Delta\alpha$ gives the color-corrected hue angle $\alpha$ for Sentinel-2 MSI. Combining the research with [20], the FUI lookup Table was defined as depicted in Table 1.

**Table 1.** FUI lookup table.

| FUI | $\alpha$ | Color Hex Code | FUI | $\alpha$ | Color Hex Code |
|-----|------|-----------|-----|--------|-----------|
| 1 | 42.27 | #7AC0F8 | 12 | 202.03 | #C4F856 |
| 2 | 50.76 | #02C7FE | 13 | 204.04 | #CBF951 |
| 3 | 64.87 | #01E2FF | 14 | 206.68 | #D8F649 |
| 4 | 80.67 | #03DAFD | 15 | 209.67 | #E1F14E |
| 5 | 104.01 | #00D9E5 | 16 | 213.32 | #EBF23F |
| 6 | 135.77 | #01D7BD | 17 | 217.89 | #F5EE3E |
| 7 | 160.08 | #00D597 | 18 | 223.39 | #FFEA39 |
| 8 | 174.86 | #00D96F | 19 | 228.28 | #FEE236 |
| 9 | 186.54 | #71E463 | 20 | 232.96 | #FCDE22 |
| 10 | 195.34 | #A5F258 | 21 | 239.00 | #FFD521 |
| 11 | 200.33 | #BBF757 | | | |

In this study, the spectral data of four ground samples were corrected using the Sentinel-2 MSI correction tone angle α. The correction tone angles α for samples 1#, 2#, 3#, and 4# were 208.78°, 208.57°, 162.83°, and 193.74°, respectively. Their corresponding FUI levels were lv15, lv15, lv8, and lv10, which demonstrated consistency with visual perception and affirmed the accuracy of the correction process. Based on these results, we implemented the aforementioned color grading process and standards into a GEE script for monitoring cyanobacterial blooms. This allowed us to automatically extract cyanobacterial blooms and assess their growth statuses through color recognition.

## 4. Results and Analysis

### 4.1. Accuracy Verification

The Jiangsu Environmental Monitoring Center in China conducts daily monitoring of cyanobacterial blooms in Lake Taihu using MODIS and NDVI in conjunction with visual interpretation. The results of this monitoring undergo consultation with the meteorological department and are publicly released at http://www.jsem.net.cn/mrygyt/thlz/ (accessed on 11 May 2023). These results, which have undergone discussions and manual corrections by multiple departments, are considered authoritative and accurately reflect the extent of cyanobacterial blooms. This study includes a linear regression analysis to assess the accuracy of the cyanobacterial bloom extraction method by comparing it with published

results from the Jiangsu Environmental Monitoring Center. Samples of cyanobacterial blooms were collected and analyzed during the period from 2019 to 2022. The results demonstrate a significant concurrence between the extracted cyanobacterial bloom areas and the published results from the Jiangsu Environmental Monitoring Center, with a linear slope exceeding 0.85 and a coefficient of determination ($R^2$) surpassing 0.7 (Figure 7).

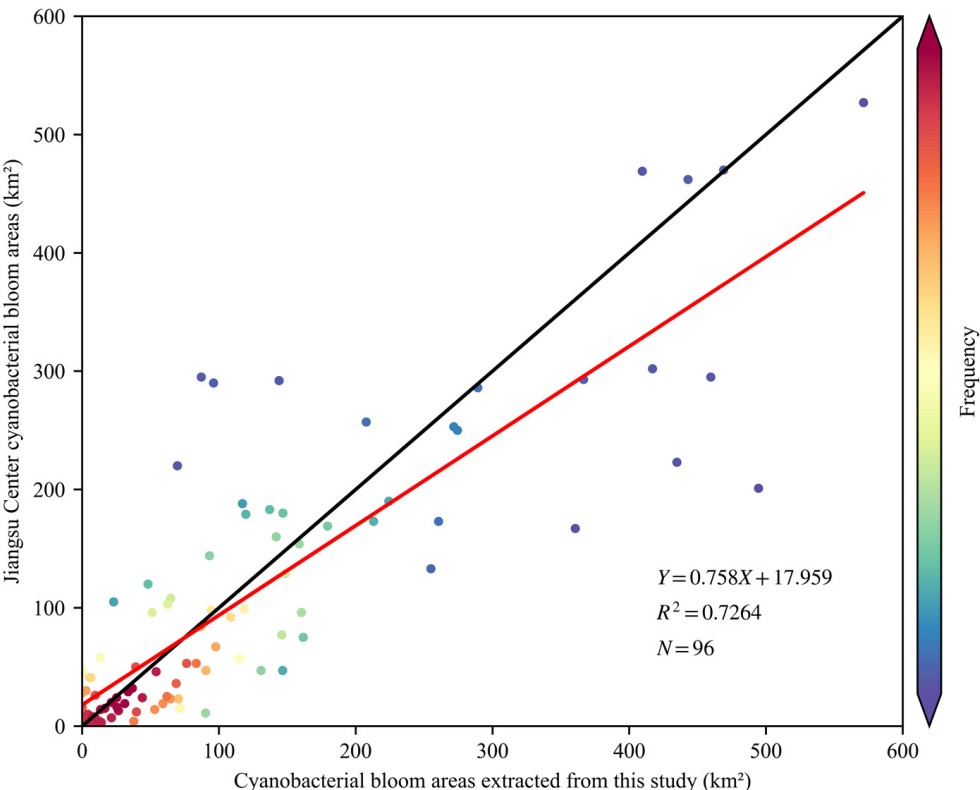

**Figure 7.** Scatter plot of our daily cyanobacteria bloom areas versus those of the Jiangsu Environmental Monitoring Center. The black line represents the 1:1 line, indicating a perfect match or proportional relationship between the datasets. The red line represents the trend line, showing the correlation between the variables.

From Figure 7, it is evident that the cyanobacterial bloom areas extracted in this study were slightly larger than the consulted results. This disparity could be attributed to two primary factors. Firstly, the utilization of Sentinel-2 MSI in this study offered superior spatial resolution compared to MODIS, enabling a more precise capture of spatial characteristics and an enhanced ability to detect cyanobacterial blooms. Secondly, due to algorithmic limitations, this study may be limited to completely exclude aquatic plants, particularly in the eastern region of Lake Taihu, where aquatic plants are occasionally misclassified partially as cyanobacterial blooms. In contrast, the consulted results employed masking and underwent manual corrections specifically for this area, resulting in a slight overestimation of cyanobacterial bloom extraction results in this study.

### 4.2. Time Series Analysis of Cyanobacterial Blooms in Lakes

Figure 8 illustrates the percentage of cyanobacterial bloom areas in four typical lakes, namely, Hulun Lake, Taihu Lake, Chaohu Lake, and Dianchi Lake, during the period from 2019 to 2022. The seasonal succession pattern is evident in Taihu Lake and Chaohu Lake, where cyanobacterial blooms rarely appear from January to April but significantly increase in frequency and size from May to December. The annual trend exhibits a consistent decline from 2019 to 2022. Chaohu Lake experiences a higher frequency of cyanobacterial blooms-free months compared to Taihu Lake, primarily spanning from November through the

subsequent May. The decreasing trend is more pronounced in comparison to Taihu Lake, especially in 2022, when large-scale cyanobacterial bloom outbreaks were observed solely in August and October. Dianchi Lake exhibits an irregular seasonal pattern, characterized by sporadic occurrences of large-scale cyanobacterial bloom outbreaks in August, November, and December 2019 and December 2020. However, since 2021, there has been almost no such outbreaks of cyanobacterial bloom. The gradual reduction in cyanobacterial blooms in Taihu Lake, Chaohu Lake, and Dianchi Lake to some extent indicates the significant progress achieved in controlling eutrophication in these three lakes. In contrast, Hulun Lake experienced mild cyanobacterial blooms outbreaks from 2019 to 2021, however, a sudden outbreak in 2022, especially during July and August. During this period, the cyanobacterial blooms covered over 30% of the lake area, with the maximum coverage reaching close to 60%. This may be due to the increasing eutrophication level of the lake, with nitrogen, phosphorus, and other nutrients gradually reaching the optimum concentrations for cyanobacterial bloom growth.

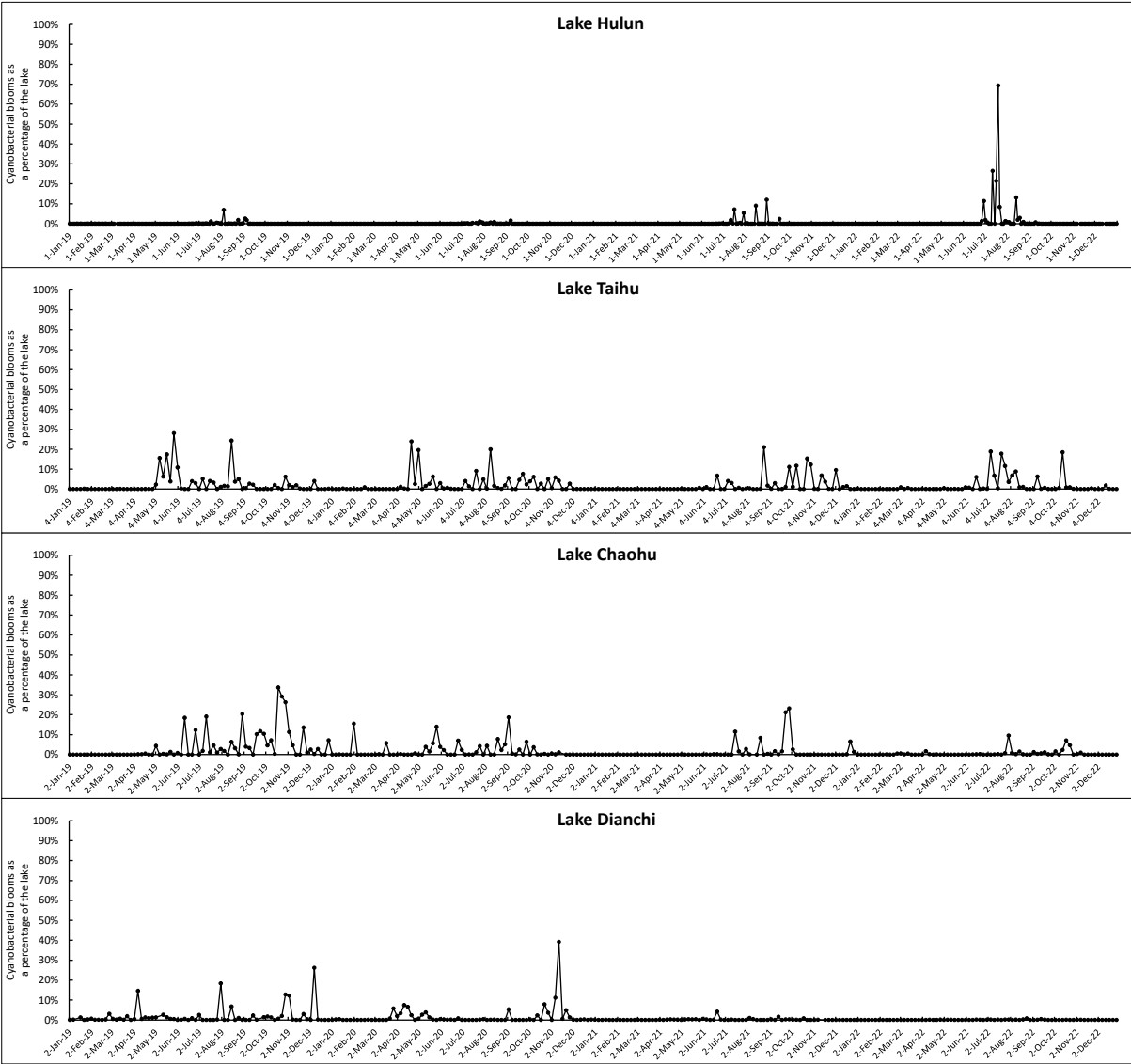

**Figure 8.** Percentage of cyanobacterial blooms in four typical eutrophic lakes from 2019 to 2022.

The maximum outbreak areas of cyanobacterial blooms during 2019–2022 are shown in Figure 9. During the period of 2019–2022, the maximum outbreak area of cyanobacterial blooms in Lake Hulun was 1423.30 km², occurring on 19 July 2022; in Lake Taihu, the

maximum outbreak area was 658.56 km², occurring on 29 May 2019; in Lake Chaohu, the maximum outbreak area was 261.55 km², occurring on 19 October 2019; in Lake Dianchi, the maximum outbreak area was 116.71 km², occurring on 12 November 2020. We developed an app (https://songting1207.users.earthengine.app/view/s2-bloom-lake-system (accessed on 11 May 2023)), which provided the aforementioned results.

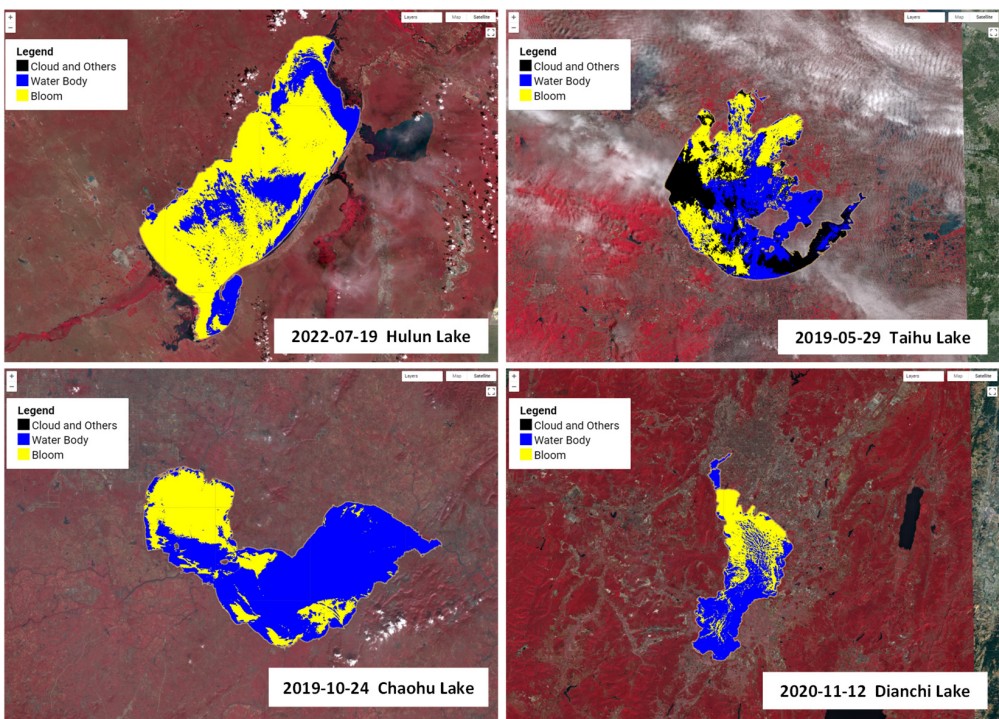

**Figure 9.** Largest cyanobacterial blooms in four typical eutrophic lakes from 2019 to 2022. (Black represents clouds and other factors, blue represents water bodies, and yellow represents cyanobacterial bloom).

By categorizing the lake area into three classifications (i.e., those affected by clouds and other factors, water bodies, and cyanobacterial blooms), it was possible to carry out a relevant analysis of the relationship between these categories. Figure 10 shows the annual average area changes in each category for the four lakes. Approximately half of the areas of Lake Hulun, Lake Taihu, and Lake Chaohu were impacted by clouds and other factors but with much less interference in Lake Dianchi, which allowed for a more accurate representation of cyanobacterial bloom extractions in relation to actual occurrences. In line with the previous discussion, the annual average values of cyanobacterial blooms in Lake Taihu, Lake Chaohu, and Lake Dianchi showed a downward trend over the years, whereas that of Lake Hulun clearly increased. The annual average values of cyanobacterial blooms in Lake Taihu, Lake Chaohu, and Lake Dianchi decreased from 53.10 km², 30.25 km², and 5.49 km² in 2019 to 37.76 km², 3.95 km², and 0.19 km² in 2022, respectively, representing decreases of 28.89%, 86.95%, and 96.53%. The annual average value of cyanobacterial blooms in Lake Hulun increased from 2.51 km² in 2019 to 23.40 km² in 2022, representing an increase of 833.03%.

### 4.3. Spatial Analysis of Cyanobacterial Blooms in Lakes

Figure 11 shows the frequency of cyanobacterial bloom outbreaks in four typical eutrophic lakes from 2019 to 2022, obtained by binarizing and overlaying the monitoring results based on Equation (10).

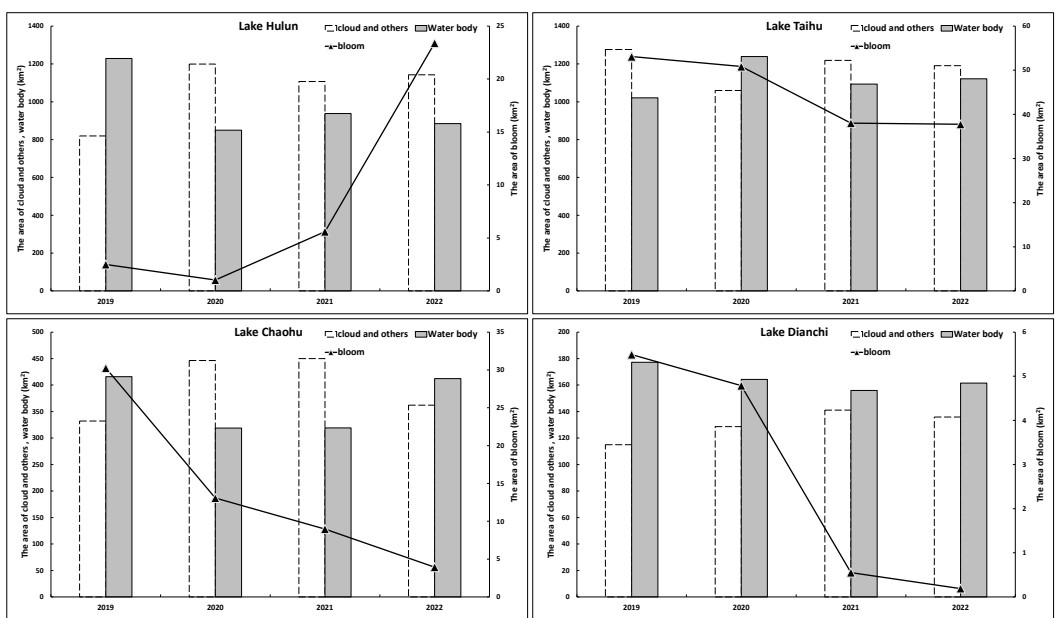

**Figure 10.** Annual average area changes in four typical eutrophic lakes in each category from 2019 to 2022.

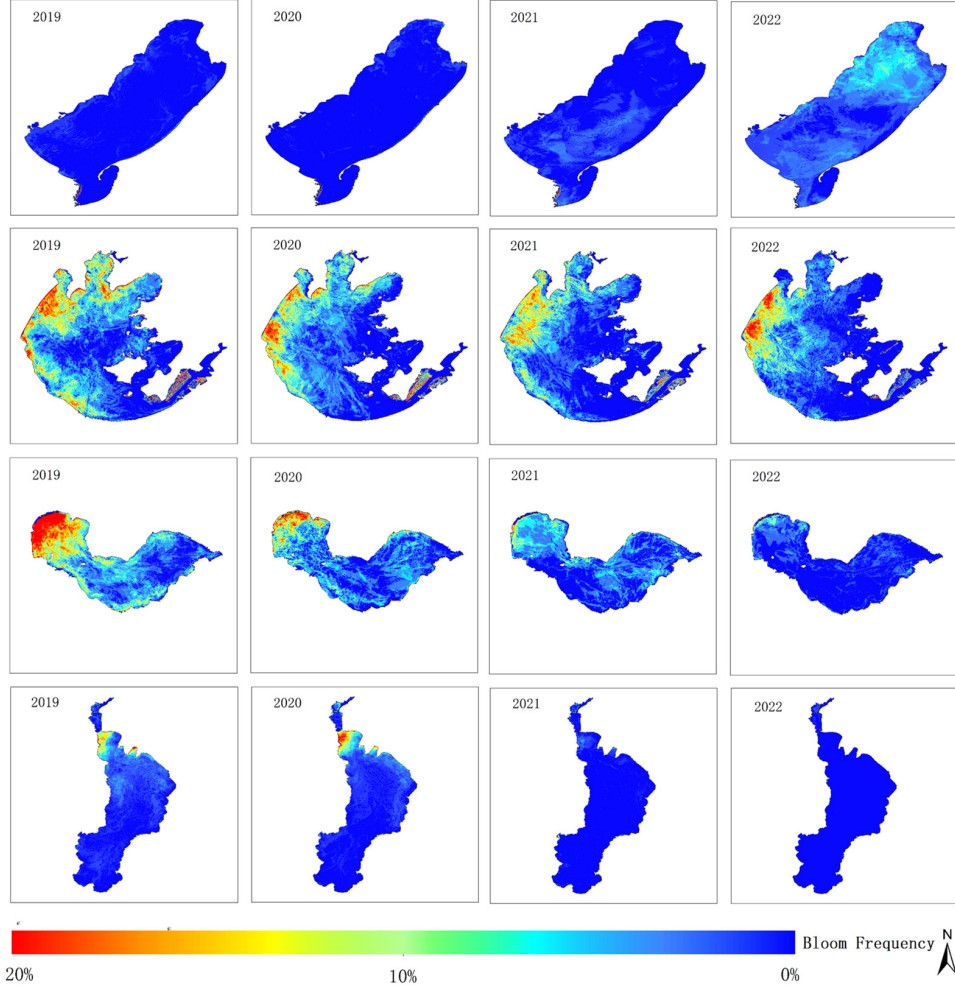

**Figure 11.** Frequency of cyanobacterial blooms in four typical eutrophic lakes from 2019 to 2022.

Cumulative annual frequencies of cyanobacterial blooms were analyzed for each lake. The blooms in Hulun Lake showed a significant increase from 2019 to 2022, with the affected area gradually extending from the lakeshore towards the center of the lake, eventually covering almost the entire lake in 2022. Conversely, there has been a gradual reduction in cyanobacterial blooms observed in Taihu Lake, Chaohu Lake, and Dianchi Lake over the years. Meiliang Lake and the western region of Taihu Lake exhibited high frequencies of cyanobacterial blooms from 2019 to 2022, whereas the frequencies in the lake centers of Gonghu Lake and Zhushan Lake decreased over time. Cyanobacterial blooms in Chaohu Lake mainly occurred in the western part of the lake, with relatively low frequencies in the central and eastern lake areas, with a decreasing trend over the years. In Dianchi Lake, high frequencies of bloom outbreaks were noted in the northern and eastern regions during 2019 and 2020, but the frequency significantly decreased during 2021 and 2022, with most of the lake not experiencing cyanobacterial blooms.

### 4.4. Lake Cyanobacterial Blooms Color and Growth Analysis

The FUI was calculated using the R, G, and B channels combination of bands B4 (665 nm), B3 (559 nm), and B2 (442 nm) to extract cyanobacterial blooms. The corrected hue angle $\alpha$ was obtained using Equation (11). Based on the color grading in Table 1, with the combination of color grading results of ground monitoring yellow-green algae and control samples, this study classified green cyanobacterial blooms as lv6–lv10, yellow-green algae blooms as lv11–lv13, and yellow algae blooms as lv14 or higher. The maximum outbreak areas of four lakes during the period from 2019 to 2022 were identified and classified based on color recognition. The color recognition results are shown in Figure 12.

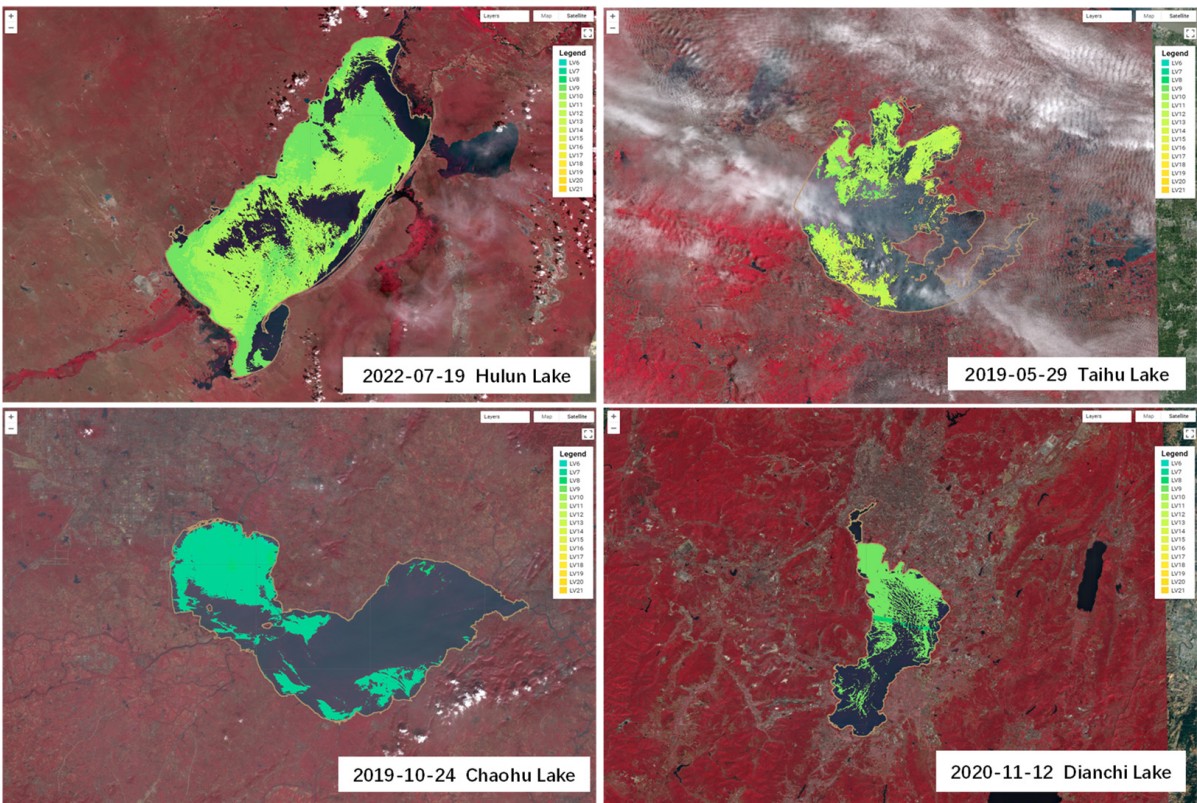

**Figure 12.** Color recognition results of the largest cyanobacterial blooms in four typical eutrophic lakes from 2019 to 2022.

The color grading results of the cyanobacterial blooms in Lake Hulun, Lake Taihu, Lake Chaohu, and Lake Dianchi from 2019 to 2022 are shown in Figure 13. The figure illustrates that the prevalence of yellow-green and yellow algae blooms was relatively high

in both Lake Taihu and Lake Dianchi. In particular, in 2019, Lake Dianchi exhibited a proportion of 24.67%, with a notably high percentage of 17.68% of yellow algae blooms. Subsequently, as the intensity of cyanobacterial blooms decreased, the proportion of yellow-green and yellow algae blooms also decreased significantly, reaching only 12.91% in 2022. The proportion of yellow-green and yellow algae blooms in Lake Taihu remained stable at a relatively high level, ranging from 15.76% to 26.24%, with the latter accounting for 3.1−5.07%. While there has been a gradual decrease in bloom intensity over time, this decline was much less pronounced than that observed in Lake Dianchi. According to data released by China's environmental monitoring authorities, the concentrations of nitrogen and phosphorus in Lake Taihu and Lake Dianchi have remained relatively low in recent years compared to previous periods. Specifically, in 2022, the total nitrogen concentration in Lake Taihu was measured at 1.25 mg/L, whereas the total phosphorus concentration was recorded at 0.062 mg/L, respectively. For Lake Dianchi, the corresponding concentrations were 2.23 mg/L for total nitrogen and 0.061 mg/L for total phosphorus. The lower levels of phosphorus concentrations in these two lakes may result in "nitrogen limitation" or "phosphorus limitation" for cyanobacterial growth, leading to more evident "yellowing" characteristics of cyanobacterial blooms.

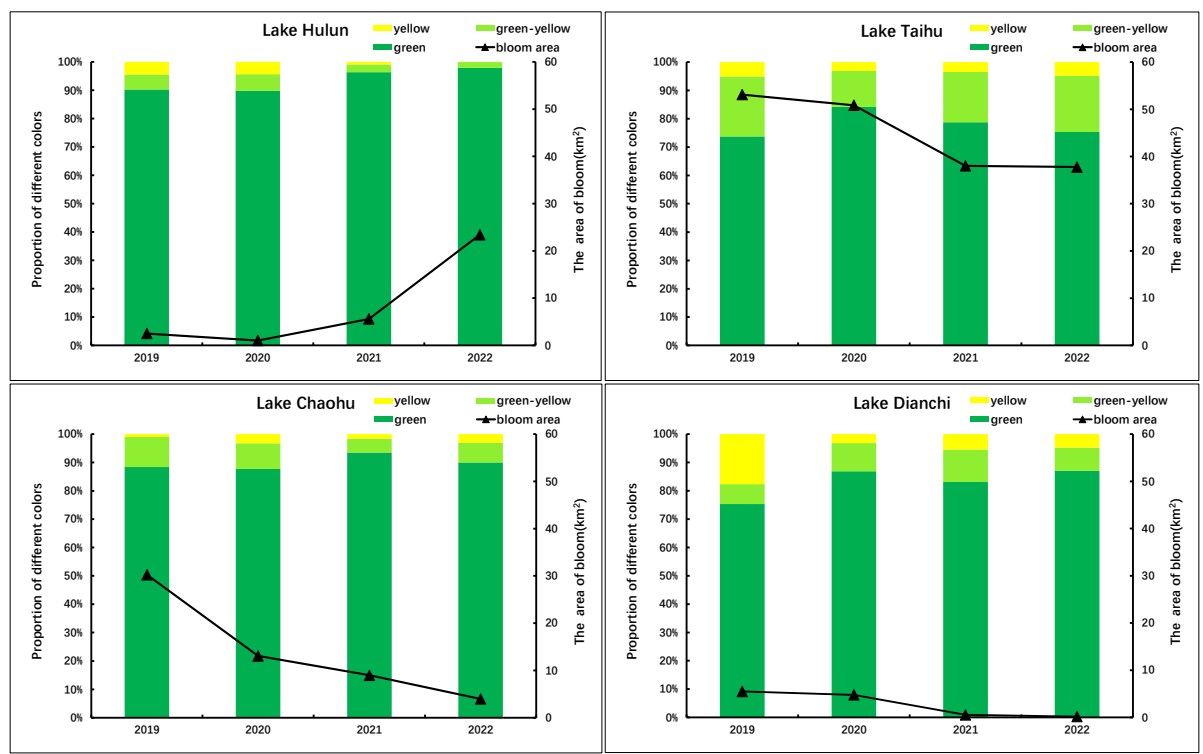

**Figure 13.** Color classification of cyanobacterial blooms in four typical eutrophic lakes from 2019 to 2022.

The proportion of yellow-green and yellow algae blooms in Lake Chaohu was relatively low, with the highest proportion in 2020 at 12.26% and the lowest proportion in 2021 at 6.52%. The intensity of algae blooms showed a gradual decreasing trend. The concentrations of total nitrogen and total phosphorus in Lake Chaohu were 1.71 mg/L and 0.082 mg/L, respectively, in 2022. Particularly, the phosphorus concentration was significantly higher than that observed in Lake Taihu and Lake Dianchi. This disparity may have resulted in a lesser degree of "yellowing" in cyanobacterial blooms in Lake Chaohu compared to the other two lakes. Furthermore, the intensity of bloom outbreaks in Lake Chaohu has gradually decreased over the years. This suggests that the nitrogen and phosphorus concentrations in Lake Chaohu may not be limiting factors for cyanobac-

terial growth, but rather other comprehensive factors have contributed to the decrease in cyanobacterial bloom intensity.

The proportion of yellow-green and yellow algae blooms in Lake Hulun was the lowest among the four lakes, with only 3.63% and 2.06% in 2021 and 2022, respectively. There was a near absence of yellow algae blooms, whereas the intensity of algae blooms exhibited an upward trend year by year. During the period from June to October 2022, the Northeast Institute of Geography and Agroecology, Chinese Academy of Sciences, conducted four sampling events in Lake Hulun. The average concentrations of nitrogen and phosphorus were 2.85 mg/L and 0.236 mg/L, respectively. These levels were significantly higher than those observed in the other three lakes, possibly due to the favorable growth conditions for cyanobacterial in Lake Hulun, including the relatively high nitrogen and phosphorus concentrations and other factors that resulted in a weak "yellowing" phenomenon of cyanobacterial blooms. The predominant color observed was green, resulting in a substantial escalation in the magnitude of cyanobacterial blooms in Lake Hulun.

## 5. Discussion

### 5.1. Limitations of Automatic Extraction Methods for Cyanobacterial Blooms

The validation results indicate that this study provides a workflow with relatively accurate precision for automated, integrated, and rapid monitoring purposes. However, there are some inevitable deviations that need consideration. Firstly, the surface reflectance data of Sentinel-2 provided on GEE may be excessively corrected compared to the Raleigh correction, which could potentially affect monitoring. Secondly, although this study used a decision tree classification based on the hue angle alpha to remove aquatic plants, it is not always feasible to completely distinguish between the hue angles of cyanobacterial blooms and aquatic plants, resulting in incomplete removal of aquatic plants. In particular, the presence of aquatic plants in the eastern part of Lake Taihu is occasionally included in the assessment of cyanobacterial blooms, leading to an overestimation of the results of cyanobacterial blooms. However, this analysis results of long time series remain unaffected by this issue, and the affected area can be masked if necessary. Another factor that affects accuracy is the adjacent effect of the ground surface. Scattered light from brighter land pixels near the water–land boundary will increase the reflectance values of the NIR band [21], making the hue angle alphas of the water pixels similar to those of the cyanobacterial blooms, resulting in misidentifications. The adjacent effect of the land surface can impact several hundred meters off the shore; however, it is imperative that the land mask remains at a sufficient size [22]. Clouds can also cause deviations, especially when processing cyanobacterial bloom pixels covered by thin clouds. These areas can be visually identified and classified into the final statistics of cyanobacterial bloom. However, the strict cloud detection method used in this study may obscure these areas, leading to an underestimation result of cyanobacterial bloom. In addition, compared to satellite sensors like MODIS (Moderate Resolution Imaging Spectroradiometer) [22] and AGRI (Advanced Geosynchronous Radiation Imager) [23], which enable high-frequency monitoring of cyanobacterial blooms, the temporal resolution of Sentinel-2's MSI is significantly inadequate for a comprehensive and complete assessment of cyanobacterial bloom occurrences. However, its higher spatial resolution (10 m) makes it effective in monitoring cyanobacterial blooms in smaller water bodies.

### 5.2. Mechanism Analysis of Yellow Cyanobacterial Blooms

Yellow cyanobacterial blooms remain a phenomenon whose underlying causes have yet to be fully comprehended. A primary point of contention is whether the cyanobacteria undergo yellowing or not. Some scientists argue that yellow cyanobacterial blooms consist of distinct algal particle groups, each with different pigment compositions, resulting in varying colors [24]. Despite the dominance of cyanobacteria in the water, the yellow color is then attributed to light refraction and the presence of red-associated diatoms. Others believe that changes in the surrounding environment can cause a reduction in the

synthesis of chlorophyll and phycocyanin in cyanobacteria, resulting in an increase in the carotenoids and xanthophylls ratio, ultimately causing yellow appearance and a yellow cyanobacterial bloom. Environmental factors that affect this include light and nutrient elements, particularly nitrogen [25,26].

Light intensity exerts a significant impact on the pigment synthesis of cyanobacterial cells [27]. Under intense sunlight, especially following a period of cloudy and rainy weather, cyanobacteria may excessively decompose water and generate an excessive amount of reactive oxygen. To prevent oxidative damage to the photosynthetic system, they synthesize carotenoid light-protective pigments, increasing the proportion of carotenoids in their pigment composition and a yellowish appearance. Conversely, the growth of cyanobacteria can be negatively impacted by a lack of sunlight or nutrients, potentially leading to their demise. Simulation experiments by Meng Zejing et al. suggested that fresh green cyanobacteria gradually degrade into yellow-green, light yellow, and white after three days of decline, during which a large amount of phycocyanin is released [28], explaining the yellowing of cyanobacteria during the decline phase. Additionally, Sidler's research demonstrated different light qualities can alter the cell phycobiliprotein components of cyanobacteria [29]. Cyanobacteria can be divided into three categories: (1) those whose phycobiliprotein components remain unchanged under different light qualities when they contain phycocyanin [30], (2) those whose chromoprotein types in phycobiliproteins change under different light qualities, such as Calothrix sp. PCC 7601 cells, where the gene encoding red-absorbing phycocyanin is replaced by the gene-encoding, blue-absorbing phycoerythrin when the cells are cultured under red light [31], and (3) those whose relative abundances of chromoproteins in phycobiliproteins change under different light qualities [32], such as *Phormidium* sp. C86 cells, where the ratio of phycocyanin, phycoerythrin, and phycoviolobilin in phycobiliproteins is 7.3:1:1 when cultured under green light but changes to 1.2:3.3:1 under red light [33].

Nutrient limitation in non-nitrogen-fixing cyanobacteria induces a series of reactions, including cessation of cellular division and significant morphological and physiological alterations such as loss of photosynthetic membranes, an increase in glycogen, and pigment degradation. Nitrogen limitation causes a reduction in chlorophyll and phycobiliprotein content, resulting in a dramatic color shift from normal blue-green to yellow-green, known as "bleaching." Phycobiliprotein undergoes extensive degradation, leading to a reduction in chlorophyll content. The degradation of phycobiliprotein may serve as a source for peptide synthesis required to adapt to nitrogen-depleted conditions. This phenomenon is also observed in certain cyanobacterial species when faced with limitations of sulfur, phosphorus, carbon, and iron nutrients. In these cases, degradation minimizes the absorption of excess excitation energy under stress conditions. Collier and Grossman have demonstrated that bleaching reactions differ under nitrogen or sulfur and phosphorus limitations [34]. In nitrogen- or sulfur-limited culture media, the degradation of phycobiliprotein occurs more rapidly and extensively than in phosphorus-limited media, indicating differential responses to nutrient limitations. In addition, Noel et al. found that the yellow cyanobacterial bloom in Rotoehu Lake may have been due to nitrogen availability being the primary limiting factor for the planktonic organism growth in the lake, as the lake contains a considerable amount of active dissolved phosphorus due to its proximity to the springs (TLp − TLn = 0.4) [35].

The yellow hue of cyanobacterial blooms may be related to the varying lifecycles of cyanobacteria during bloom periods. Similar to plant leaves, cyanobacteria undergo a seasonal lifecycle and enter a senescence period as solar radiation decreases in autumn and winter. The degradation rate of the primary chlorophyll within the cell gradually increases, leading to a corresponding increase in the proportion of other pigments, including yellow. Furthermore, external factors such as nutrient concentration and weather conditions can induce an accelerated growth cycle and senescence within a relatively short period of time due to nutrient limitations causing bloom extinction. During this stage, on the one hand, the synthesis of chlorophyll a and phycocyanin, which contains four nitrogen

molecules, is affected by nitrogen limitations. On the other hand, chlorophyll a is relatively photosensitive and prone to decomposition, whereas carotenoids are relatively stable. As a result, there is a decrease in cell chlorophyll a concentration and an increase in carotenoid content, leading to the yellow color of the cyanobacterial blooms.

Overall, a deficiency of soluble nutrients such as nitrogen, phosphorus, and sulfur, as well as the production of photoprotective pigments due to alterations in light cycle and light stress, can induce physiological and metabolic responses in cyanobacteria, leading to changes in pigment composition and visual color perception. Based on the results of color recognition of cyanobacterial blooms in four typical eutrophic lakes in China, each lake exhibits unique characteristics in terms of the yellowing phenomena during cyanobacterial blooms. In addition to the impact of intense solar radiation and the lifecycle of cyanobacterial blooms, regional variations are reflected in nutrient concentration. Different nutrient statuses lead to distinct "yellowing" phenomena and growth states of cyanobacterial blooms in each lake. The yellowing of cyanobacterial blooms induced by nitrogen or phosphorus limitation can, to a certain extent, predict changes in the intensity of cyanobacterial blooms in the lakes of the study area.

*5.3. Future Outlook*

Inland water bodies possess complex inherent optical properties that make it difficult to apply current methods across different regions. Nevertheless, this study implemented relevant processing procedures into the GEE cloud platform, achieving acceptable accuracy and obtaining spatial and temporal patterns of cyanobacterial blooms in eutrophic lakes. We believe that the workflow developed in this study can help practitioners apply it to other lakes, especially those of small-to-medium size. Notably, the cyanobacterial bloom color recognition module, in particular, can assess the nutrient status of lakes and forecast changes in bloom occurrence.

In future research, it is recommended to incorporate diverse atmospheric correction algorithms and cyanobacterial bloom extraction methods for specific inland water bodies. It should be noted that the cyanobacterial blooms extracted in this study only represent the water body's surface distribution and cannot be used to quantify cyanobacterial biomass. Therefore, relevant remote sensing inversion algorithms may be embedded in the workflow in the future to account for the vertical distribution of cyanobacterial biomass within the water column. In addition, deep learning algorithms will improve the recognition and processing for thin clouds, enabling better separate cyanobacterial blooms from aquatic plants. Continued improvements in the cyanobacterial bloom monitoring and color recognition algorithms and taking advantage of the cloud platform's big data will provide technical support for cyanobacterial bloom early warning monitoring and prevention and control management.

## 6. Conclusions

This study developed an operational workflow for monitoring cyanobacterial blooms using the GEE platform and Sentinel-2 MSI data. Over 7000 Sentinel-2 images were processed to analyze the spatiotemporal distribution of cyanobacterial blooms in four typical eutrophic lakes in China: Lake Hulun, Lake Taihu, Lake Chaohu, and Lake Dianchi. The FUI was used to identify and evaluate the color of cyanobacterial blooms. The study analyzed sources of inaccuracy in the workflow, including cyanobacterial extraction algorithms, land adjacency effects, and cloud impacts. Despite these sources of error, the workflow achieved acceptable accuracy when compared to cyanobacterial monitoring results from the Jiangsu Environmental Monitoring Center. The study demonstrated that the implementation of the workflow could improve the efficiency of automated monitoring of cyanobacterial blooms in inland water bodies while revealing the spatiotemporal trends in cyanobacterial blooms across the four lakes as well. Cyanobacterial blooms in Lake Taihu, Lake Chaohu, and Lake Dianchi showed decreasing trends over time, whereas those in Lake Hulun increased, reaching the most severe outbreak in 2022. Additionally, the study

revealed that the color of cyanobacterial blooms varied among these four lakes. Specifically, yellow-green and yellow blooms were more prevalent in Lake Taihu and Lake Dianchi compared to Lake Chaohu and Lake Hulun. The geographical variation in the yellowing status of cyanobacterial blooms was found to be primarily associated with nutrient concentration, specifically "nitrogen limitation" or "phosphorus limitation", which contribute to the occurrence of the cyanobacterial bloom discolorations. This discovery can help predict changes in the intensity of cyanobacterial blooms within study areas.

**Author Contributions:** T.S.: Conceptualization, Data curation, Formal analysis, Methodology, Visualization, Roles/Writing—original draft; G.L.: Project administration, Supervision, Writing—review and editing; H.Z.: Software, Investigation; Y.F.: Investigation, Resources; F.Y.: Software, Visualization; J.Z.: Funding acquisition, Project administration, Supervision, Writing—review and editing. All authors have read and agreed to the published version of the manuscript.

**Funding:** The research was jointly supported by [the Ecological and Environmental Research Achievements Transformation and Promotion Project of Jiangsu Province] grant number [2022012]; [National Key Research and Development Program of China] grant number [2021YFB3901101].

**Data Availability Statement:** The sentinel-2 dataset is available in the data archive of Google Earth Engine (https://developers.google.com/earth-engine/datasets (accessed on 11 May 2023)). The code for the automatic algal bloom detection algorithm can be accessed at (https://songting1207.users.earthengine.app/view/s2-bloom-lake-system (accessed on 11 May 2023)), and the code for other analysis of this study is available from the corresponding author upon reasonable request.

**Acknowledgments:** Data produced and analyzed in this paper were generated in collaboration with Northeast Institute of Geography and Agroecology, Chinese Academy of Sciences.

**Conflicts of Interest:** The authors declare no conflict of interest.

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
