# Peer review of "Lake Cyanobacterial Bloom Color Recognition and Spatiotemporal Monitoring with Google Earth Engine and the Forel-Ule Index"

_remotesensing, doi:10.3390/rs15143541_

Round 1

Reviewer 1 Report

Comments

Point 1: Revise English.  

 Point 2: Section M&M. subsection 2.1. Study area

 I suggest placing the lakes in a separate figure, integrated in fig 1 they seem to be in the sea. See Figure 5.

In this sense it is possible to see:

1. Presence of tributaries (away from their influence).

2. Presence of industrial effluents or urban discharges.

3. Depth.

Topography would be a plus to be added

 Added brief description of Morphometric parameters in each study lake. example, Long maximum in km, Maximum width, Superficial area, Medium depth, Volume m3, etc

Point 3: Figure 9 is not clearly visible. remove decimals from %. Put the same scale (100%) on all figures to better see the trend.

Point 4: Figure 10, 11, 13 is not clearly visible.

 Point 5: Figure 14 remove decimals from proportion of different colors % and Put the same scale for showed the area of bloom.

 Point 6. The discussion should be enriched with more refs. and specific section on the limitations of this methodology.  

I suggest reviewing the following reference: https://www.mdpi.com/2073-4441/15/12/2165

Reviewer 2 Report

The authors' work on the distribution of cyanobacterial blooms and the color differences of cyanobacterial blooms in four typical lakes in China is of some research value, but there are several issues that need to be clarified by the authors:

1) The authors believe that the use of FUI index and Sentinel-2 MSI is suitable for monitoring cyanobacterial blooms in small and medium-sized lakes, especially smaller water bodies, but the lakes selected by the authors are all large lakes, what is the value of this method for the authors to sample?

2) The authors sampled a large number of subjective judgments and lacked empirical studies when analyzing the controlling factors for the occurrence of the four cyanobacterial blooms. In particular, the authors suggest that the yellowing state of cyanobacterial blooms is more related to nutrient concentrations, where "nitrogen limitation" or "phosphorus limitation" causes the "yellowing" of cyanobacterial blooms. The authors need additional evidence.

3) In discussing the accuracy of the results, the authors refer to the area data of cyanobacterial blooms obtained by the Jiangsu Environmental Monitoring Center based on MODIS data and NDVI. On the one hand, one set of remote sensing products is validated with another set of remote sensing products, and the accuracy of the reference data products should be stated first. On the other hand, it may be more convincing to the readers if the occurrence area is used for validation.

4) Most of Figures in the text are in small font, which affects the reading. There are many first-person expressions used in the text, which could be appropriately revised.

Reviewer 4 Report

Line 50, sentence "..blooms are been observed in .." please correct the grammar.

Line 52, sentence "In 2000, a eutrophic lake, Lake Rotoehu in New..", please rephrase the sentence to make it grammatically accurate. 

Line 53 "Some people believe that this color change..." this is a very unprofessional statement that has to be altered or removed.

Lines 61 to 63 "In China, such blooms are very common as in freshwater lakes as the rapid development of the economy has led to their eutrophication" should also be altered. 

Please change the title of Fig.1 "Locations and morphology of lakes." as it is misleading. The figure provides the locations and shapes of the lakes.

Line 148, leave out the word "analysis"

Please provide additional explanation as from Fig. 3 it is not clear does the algorithm recognize by itself if there is algal bloom or is this step left to the user? 

The formatting of the equation is not uniform throughout the manuscript.

The same is valid for the figures.

Line 262, there is a typo, "..blooms bloom..".

Considering the numerous issues found throughout the paper, I would advise the Authors to implement a thorough revision before their next submission.  The listed issues are only some of the errors found within the paper.

Is referred app available only in Chinese? If so please provide a translation to make it approachable to the readers, or at least the reviewers. 

There are numerous language errors throughout the Manuscript suggesting it would be beneficial for the Authors to conduct a thorough revision of their paper before their next submission. Although these are not severe issues and the language is quite satisfying, the number of errors is notable and is the main reason I mentioned it here.

Round 2

Reviewer 1 Report

The authors have satisfactorily answered the suggestions sent. They should only incorporate in the figure "study area" the coordinates (latitude and longitude).
If possible, I suggest to take out from Figure 10 the daily bloom statistics, it really does not differ much. Place it better on the right side of the figure or in supporting information.
Likewise for figure 13

Author Response

Thank you. As you suggested, we have modified Figure 1, Figure 2, Figure 5, Figure 10, and Figure 13.

Reviewer 4 Report

Thank you for the implemented improvements.

Author Response

Thank you very much for your review of this article